# Antimicrobial Biomaterials for the Healing of Infected Bone Tissue: A Systematic Review of Microtomographic Data on Experimental Animal Models

**DOI:** 10.3390/jfb13040193

**Published:** 2022-10-18

**Authors:** Lorena Castro Mariano, Maria Helena Raposo Fernandes, Pedro Sousa Gomes

**Affiliations:** 1BoneLab—Laboratory for Bone Metabolism and Regeneration, Faculty of Dental Medicine, University of Porto, 4200-393 Porto, Portugal; 2LAQV/REQUIMTE, Faculty of Dental Medicine, University of Porto, 4200-393 Porto, Portugal

**Keywords:** antibacterial biomaterials, animal models, bone healing, microtomography

## Abstract

Bone tissue infection is a major clinical challenge with high morbidity and a significant healthcare burden. Therapeutic approaches are usually based on systemic antibacterial therapies, despite the potential adverse effects associated with antibiotic resistance, persistent and opportunistic infections, hypersensitivity, and toxicity issues. Most recently, tissue engineering strategies, embracing local delivery systems and antibacterial biomaterials, have emerged as a promising alternative to systemic treatments. Despite the reported efficacy in managing bacterial infection, little is known regarding the outcomes of these devices on the bone healing process. Accordingly, this systematic review aims, for the first time, to characterize the efficacy of antibacterial biomaterials/tissue engineering constructs on the healing process of the infected bone within experimental animal models and upon microtomographic characterization. Briefly, a systematic evaluation of pre-clinical studies was performed according to the PRISMA guidelines, further complemented with bias analysis and methodological quality assessments. Data reported a significant improvement in the healing of the infected bone when an antibacterial construct was implanted, compared with the control—construct devoid of antibacterial activity, particularly at longer time points. Furthermore, considering the assessment of bias, most included studies revealed an inadequate reporting methodology, which may lead to an unclear or high risk of bias and directly hinder future studies.

## 1. Introduction

Bone is a complex and mineralized connective tissue organized by structure that, upon disturbance of the physiological equilibrium, requires a well-orchestrated healing process to achieve full regeneration, involving inflammatory, reparative, and remodeling phases [1]. Overall, bone displays a high intrinsic regenerative ability after an injury or disease. Nevertheless, associated local and/or systemic conditions, extensive damage or resection, or tissue infection seem to compromise the tissue’s regenerative potential [2]. Regarding the latter, *S. aureus* is the predominant etiologic agent of bone tissue infection [3]. Upon adhesion and proliferation, bacterial agents lead to an inflammatory activation combined with increased osteoclastogenesis and osteoblastic death [4], a process regulated by an elevated level of cytokines and distinct inflammatory mediators that converge to extensive bone loss and tissue destruction [5]. At the same time, an infection can further contribute to tissue thrombosis through the production of coagulases, which in conjunction with the local activation of the inflammatory response, may culminate in bone necrosis [6,7]. 

Clinical management of bone tissue infection broadly relies on systemic therapy with antibacterial agents, aiming for the reduction of the bacterial burden, consequently priming bone healing [5]. Nonetheless, systemic therapies have been associated with potential adverse effects, such as antibiotic resistance and the spread of drug-resistant bacteria, opportunistic infections, hypersensitivity, and severe cutaneous adverse reactions, gastrointestinal disturbances, toxicity, among others [8]. Accordingly, new advances in the management of bone tissue infection rely on the implantation of antibacterial biomaterials/tissue engineering constructs that are either intrinsically antibacterial or allow the local delivery of an antibacterial agent, thus targeting bacteria locally and surpassing current limitations associated with systemic therapies [9,10,11]. Additionally, these innovative systems are developed to simultaneously enhance the bone healing process by priming the functionality of osteoblastic populations, enhancing the metabolic activity and/or cell differentiation to heighten tissue healing/regeneration [10,11]. Despite the relevance of these approaches, little information is available regarding the efficacy of these innovative strategies on clinical or pre-clinical models. Therefore, this systematic review aims to evaluate the efficacy of innovative antibacterial biomaterials/tissue engineering constructs on the healing process of infected bone tissue within experimental animal models. Given the very high resolution and three-dimensional histomorphometric capabilities of microtomography—currently the gold-standard method for assessing bone microarchitecture [12,13]—only studies disclosing this evaluation methodology were included in the review. 

## 2. Methods

### 2.1. Protocol and Registration

A review protocol was developed based on the Preferred Reporting Items for Systematic reviews and Meta-Analyses (PRISMA) guidelines [14], and adapted from the structure provided in the Systematic Review Protocol for Animal Intervention Studies [15]. This protocol was registered in PROSPERO with registration number: CRD42020197148.

### 2.2. Eligibility Criteria

The focused question of this SR was formulated using the acronym PICOS (Population, Intervention, Comparison, Outcomes, Studies), of which: Population—experimental animal models of bone tissue infection; Intervention—orthotopic bone implantation of scaffold/tissue engineering construct with antibacterial activity; Comparison—orthotopic bone implantation of scaffold/tissue engineering construct devoid of antibacterial activity; Outcome—quantitative assessment of the bone healing process by microtomographic analysis; Studies—studies with a defined experimental and control group. No restrictions regarding the time or language of publication were applied.

The following exclusion criteria were applied: (1) absence of a control group; (2) studies that did not present the control group with a similar scaffold/tissue engineering construct devoid of the antibacterial strategy; (3) studies lacking quantitative assessment of the newly formed bone tissue by microtomographic analysis; (4) in vitro or ex vivo studies, human clinical trials, observational studies, case reports, systematic or narrative literature reviews, meta-analyses, pilot studies, protocols, and short communications.

### 2.3. Information Sources and Search Strategy

Electronic databases of Cochrane Library, PubMed, Scopus, and Web of Science were searched for relevant literature. Three grey literature databases were also consulted, including Google Scholar, Open Grey, and ProQuest. In addition, the reference list of the included studies was also screened to find additional eligible articles not retrieved by the electronic search. The last search was performed in August 2022. Computer software was used for reference management and duplicate removal (EndNote X7^®^, Thomson Reuters, Philadelphia, PA, USA).

### 2.4. Study Selection and Data Extraction 

Two independent reviewers (L.M. and P.G) screened each article via two phases: (1) title/abstract screening and (2) full-text screening. Uncertainty in the determination of eligibility was resolved by discussion with a third investigator (M.F). At the final stage, full-text manuscripts were screened based on the inclusion criteria to confirm the eligibility of each study (Figure 1). 

Data was preferentially extracted from result tables in the selected articles. If the data was not presented in a table format, a detailed search was carried out in the results section. If the data was not available or in the case in which more information was needed, the corresponding authors were contacted via the electronic address (maximum 2 times). The following study characteristics were required: author(s), year of publication, animal characteristics (i.e., species, age, size, sex, weight), bacteria strain for the infection model, amount of bacteria and inoculation method, defect location, number of animals in each group, type of scaffold/tissue engineering construct, antibacterial strategy and quantitative assessment of new bone formation.

### 2.5. Risk of Bias Assessment 

The risk of studies’ bias was assessed using SYRCLE’s Risk of Bias tool [16]. This tool contains 10 items to assign a judgment. The following criteria were used to determine the bias level of each study: (1) selection bias (random sequence generation, allocation concealment, and baseline characteristics); (2) performance bias (blinding of personnel); (3) detection bias (blinding of outcome assessment); (4) attrition bias (incomplete outcome data); (5) reporting bias (selective outcome reporting); and (6) other sources of bias. The quality of each study was evaluated by the judgment of “Yes,” “No,” or “Unclear.” “Yes” judgment indicated a low risk of bias; “No” judgment indicated a high risk of bias; “Unclear” judgment was suggested if insufficient details had been reported to evaluate the risk of bias properly. The risk of bias analysis was performed using the software RevMan Version 5.3 (The Nordic Cochrane Centre, The Cochrane Collaboration, Copenhagen, 2014).

### 2.6. Summary Measures

Due to the attained heterogeneity regarding the study design of the included studies, such as different animal models, defect locations, diversity of biomaterials/tissue engineering constructs, antibacterial strategies, outcome measures, and follow-up time, no meta-analysis could be performed. 

## 3. Results

### 3.1. Study Selection

A total of 768 articles were initially identified in the electronic databases. After removing duplicates, 475 studies remained. A total of 438 studies were excluded based on title and abstract, which left 37 studies for full-text screening. Of these, 10 studies were included in the synthesis. A list of studies excluded after full-text read, with reasons for exclusion, is reported in the Appendix A section. 

### 3.2. Study Characteristics

The characteristics of all included studies are presented in Table 1. All the ten included studies [17,18,19,20,21,22,23,24,25,26] were in vivo animal studies published in the English language, between 2014 and 2021. Regarding the selection of the animal model, only one study used a large animal, female sheep [17]. Small animal models were predominant and included five studies with rats [18,21,22,25,26] and four studies with rabbits [19,20,23,24]. Among the rabbit models, three studies used male rabbits [19,23,24], and one study used female rabbits [20]; while regarding the use of the rat model, three studies used male rats [21,22,25], one study used both male and female animals [26]. One study did not report gender selection [18]. Regarding the follow-up time, it ranged differently according to the selected species—the assessment of the sheep model ranged from 2 to 13 weeks [17], rabbits evaluation ranged from 3 to 12 weeks [20,23,24,27], and rats were characterized from 2 to 24 weeks [18,21,22,26,28]. The sample size ranged from 6 to 70 animals.

Three different bone sites were chosen for the establishment of the defect and infection model, including the femur [17,21,25], tibia [19,20,22,23,24,26], and calvaria [18]. Regarding the selection of the bacterial agent, all studies used *S. aureus* [17,18,19,20,21,22,23,24,25,26,29]; among them, one study used a methicillin-resistant Staphylococcus Aureus (MRSA) species [19], and another study used methicillin-sensitive Staphylococcus Aureus (MSSA) species [21]. The inoculum ranged from 105 to 108 CFUs [17,18,19,20,21,22,23,24,25,26]. Tissue infection was validated by the count of the CFUs [17,18,19,20,21,22,23,24,25,26], supplemented by the assessment of serum inflammatory indices—as the C-reactive protein (CRP) and white blood cell (WBC) count [24], observation of bacterial cells upon histological preparation [25], and gene expression assessment on the granulation tissue [18].

A variety of scaffolds were used in the included studies. The most prevalent were polymeric scaffolds, as the tri-block copolymer consisting of two poly(L-lactide) (PLLA) layers interspaced by poly(ethylene glycol) (PEG) [18], polycaprolactone (PCL)/auto-polymerized poly-dopamine (PDA) [20], poly(D,L-lactic acid-co-glycolic acid) (PLGA) and polyethylene glycol (PEG) scaffolds [17]; and a natural polymer as a chitosan (CS)-based hydrogel [24]; but also calcium phosphate-calcium sulfate (CaP/CaS) composites [22]; metal-based Ti6Al4V scaffolds (TSs) [19]; and lastly, composite scaffolds such as the ABVF—a polymeric matrix (synthesized by PLGA, PEG, and PCL) containing bioactive glass (BG) [26]; a hydroxyapatite collagen (HAp/Col) structure [21] and a hydroxyapatite (HA)/polyurethane (PU) composite [23,25].

The antibacterial strategy varied among the studies, with a wide range of selected drugs (e.g., gentamicin, vancomycin, clindamycin, and cefotiam, either isolated or associated) [17,19,21,22,24,26]. Additionally, four studies used silver (Ag) [18,20,23,25] in distinct formulations, with one study conjoining Ag with tannin (T) [25].

### 3.3. Quantitative Assessment of Bone Healing

Within the included studies, the bone healing process was evaluated through microtomographic analysis (μCT), allowing for the quantitative assessment of distinct bone morphometric indexes. The bone volume fraction (BV/TV), the ratio of the segmented bone volume to the total volume of the region of interest [12], was reported in five studies (Table 2). Higher values were consistently reported within the groups implanted with the antibacterial strategy, with significant differences being attained at intermediate to long time points. Briefly, in the earliest stage of new bone formation (three weeks), Zhang et al. [23] did not find a significant difference between the experimental group and the control, while at six weeks of follow-up, significantly increased values of new bone formation were attained in the antibacterial biomaterial group. Similarly, a study that used vancomycin as the antibacterial strategy also showed higher levels at six weeks post-implantation [22]. This trend of increased bone formation with time was even more evident at eight weeks of follow-up in the Ag-NPs group, as compared with the control [20]; while a similar trend was attained upon the implantation of a chitosan-based-thermosensitive hydrogel loaded with vancomycin-nanoparticles [24]. Equivalently, at the twelve weeks post-operative period, in a rabbit model, increased and significant differences were attained between the groups implanted with the antibacterial strategy and the control [23], further increasing at the twenty-four weeks post-operative time point [18].

Regarding bone mineral density (BMD) (Table 3), which evaluates the volumetric density of calcium hydroxyapatite in biological tissues, three studies reported significantly different values between the experimental and control groups [18,24,25]. At four weeks of follow-up, a study conducted in a rat model revealed significantly higher BMD levels in the group treated with Ag/tannin compared with the control group. This result was also evidenced at eight weeks of follow-up [25]. Additionally, at eight weeks, a rabbit model treated with VCM-NPs/Gel showed significant differences between the groups, but the numerical results were not specified [24]. A rat model treated with Ag and tannin showed that BMD value significantly increased at week twelve of follow-up compared with the control group [25]. In contrast, in a longer follow-up period of twenty-four weeks, in a rat model, significant higher differences were attained between the experimental group and the control [18].

Other studies calculated the reduction of the bone defect area or volume. At four weeks post-operation, Egawa et al. [21] evaluated the bone defect area and showed a significantly decreased level in the treatment group of vancomycin compared with the control group. The same trend was noted in the other antibiotic agent groups (cefotiam group), but differences were not significant [21]. Regarding cortical bone destruction (Table 4), vancomycin and cefotiam groups induced a decrease in cortical bone destruction one week after the surgical procedure, but only the vancomycin group revealed significant results (0.45 ± 0.04 mm). At two weeks post-operation, both antimicrobial agents’ groups showed a pronounced decrease in cortical bone destruction (vancomycin group: 0.51 ± 0.07 mm, cefotiam group: 0.50 ± 0.04 mm, and control group: 0.18 ± 0.07 mm) [21]. While bone area evaluation was reported in only one study [22] (Table 5)—at six weeks of treatment with vancomycin, a rat model showed an identical value of the bone area in the antibacterial biomaterial and control group (1.46 mm2) [22]. Other studies addressed the bone fill of the defect. In one study, at 13 weeks post-operation, the antibacterial biomaterial group of the sheep model showed a bone defect fill of 53.8 ± 17.2%, while the antibiotic-free biomaterial group was found to be higher—68.4 ± 13.0%, despite the absence of significant differences [17]. In another study, no quantitative data were presented, but a qualitative morphological analysis revealed no significant difference in the bone fill of defects between the antibacterial biomaterial group and control at two weeks post-operation. At eight weeks of follow-up, a healing bone without signs of infection was reported in the rat tibia within the treatment group, being progressively filled by immature cancellous and cortical bone [26].

Distinct studies presented histomorphometric data from the trabecular bone analysis. Two studies reported the trabecular number (Tb.N) (Table 6), the number of trabeculae per unit of length of the regenerated bone, in a rabbit model, using Ag-NPs as an antibacterial biomaterial [20,23]. In one study, at the four times points evaluated, three, six, eight, and twelve weeks, higher quantitative measures were reported in the antibacterial biomaterial group, despite that only at six weeks of follow-up Tb.N revealed significantly higher levels [23]. In another study, at eight weeks, significantly higher levels were attained in the groups that contained Ag-NPs, as compared with the control group [20]. Throughout the twelve weeks of follow-up, Tb.N continued to increase, and there were no significant differences between the groups with different dosages of Ag (Ag 3% and Ag 10%) [23]. Trabecular thickness (Tb.Th), the mean thickness of the attained trabeculae, was also evaluated (Table 7). A study showed a significant difference between the antibacterial biomaterial group (Ag 3%) and control group at three, six, and twelve weeks post-operation (antibacterial biomaterial group: 0.142 ± 0.031 μm at three weeks, 0.327 ± 0.040 μm at six weeks, and 0.140 ±0.095 μm at twelve weeks; Control group 0.029 ± 0.001 μm at three weeks, 0.067 ± 0.026 μm at six weeks, and 0.157 ± 0.065 μm at 12 weeks) [23]. Although another study reported the assessment of Tb.Th, quantitative data was not presented [17].

Lastly, only one study addressed trabecular spacing (Tb.Sp), which is the mean distance between the trabeculae (Table 8). At three weeks, there were no significant differences between any of the groups. At six weeks (Ag 3%: 0.351 ± 0.099 μm and Control group: 1.245 ± 0.276 μm) and 12 weeks (Ag 3% group: 0.250 ±0.022 μm and Control group: 0.507 ± 0.122 μm) after the surgical procedure, significant differences were observed between the antibacterial biomaterial group of Ag 3% and the control. In another group of the antibacterial biomaterial (Ag 10%), there was a significant difference between this group and the control group at twelve weeks of follow-up (Ag 10% group: 0.218 ± 0.055 μm and Control group: 0.507 ± 0.122 μm), despite the absence of differences between both experimental groups [23].

### 3.4. Risk of Bias Assessment

The risk of bias assessment for each study is summarized in Table 9 and Figure 2. No studies fulfilled all the methodological criteria analyzed. The included studies in this SR contained insufficient reporting of the experimental details. Most of the domains were categorized as being with an unclear risk of bias. Regarding the selection bias, the sequence generation and baseline characteristics were not reported in all studies [17,18,19,20,21,22,23,24,25,26]. It was not clear if half of the studies reported information regarding allocation concealment [21,23,24,25,26], and the other five studies reported that allocation was properly concealed [17,18,19,20,22]. Six studies did not report random housing [18,19,20,23,25,26]. None of the studies reported whether researchers were blinded and whether outcome assessors were blinded [17,18,19,20,21,22,23,24,25,26]. In addition, none of the studies reported random outcome assessment for detection bias [17,18,19,20,21,22,23,24,25,26]. Five studies showed incomplete data outcomes [18,21,23,24,25]. Regarding reporting bias, four studies did not show a selective outcome reporting [18,19,20,24]. Additionally, four studies presented other estimated potential risks of bias, such as the lack of reported results of selected time-points [18], differences in selected time-point between the methodology and the results sections [20], the comparison of acrylic and ceramic materials(22) and the lack of quantitative data reporting regarding μCT analysis(26).

## 4. Discussion

To the best of the authors’ knowledge, this is the first systematic review to evaluate the efficacy of antibacterial biomaterials/tissue engineering strategies on the enhancement of bone healing in experimental bone tissue infection models. In this study, we systematically reviewed ten animal studies that qualified for the pre-established inclusion criteria. To determine if the antibacterial strategy effectively contributed to the enhanced bone-healing outcome in infected conditions, an appropriate control group devoid of the antibacterial approach should have been characterized. Unfortunately, due to the high degree of heterogeneity among the experimental protocols, as for animal species, study design, different bone implantation locations, type and composition of scaffolds and antibacterial strategies, and mainly the measurement methods and outcome criteria, it was not possible to conduct a meta-analysis. Future studies should leverage insights obtained from our analysis to design preclinical studies with more homogeneous protocols, enabling comparison and enhancing background data quality for the optimized planning of prospective clinical studies.

Overall, our results showed that for the bone healing in infected models, the biomaterials/scaffolds containing antibacterial strategies yielded increased histomorphometric indexes associated with bone formation, compared with the implantation of scaffolds without antibacterial strategies, particularly at longer time points of evaluation. These systems are expected to accelerate the process of bone healing in the presence of tissue infection, reducing the bacterial load and further increasing the new bone formation process. From a clinical point of view, these strategies are expected to contribute to the eradication of bacterial infection, assisting in the enhancement of the healing process and further reducing tissue morbidity.

### 4.1. Animal Models

Animal models are extremely useful in the assessment of biosafety and biofunctionality of innovative therapeutic approaches, allowing a detailed understanding of whether these strategies are appropriate for the translation from bench to bedside. Although no ideal animal model exists for the assessment of the efficacy of biomaterials/tissue engineering constructs for the healing of the infected bone tissue, a similarity to molecular, cellular, structural, and mechanical features akin to human bone, a translational infection methodology and a size adequate to the requirements of the experimental design, are greatly envisaged [30,31].

In the present systematic review, different animal models were identified and reported. Among them, the most used were rats (five), rabbits (four), and sheep (one). Rats and rabbits are relatively inexpensive and easy to handle, manipulate and maintain [30], possibly substantiating the verified preference. Bone micro- and macrostructure, as well as the bone healing process, vary among species. Furthermore, biological processes can also vary within the same species, depending on the age, gender, general condition of systemic health, and biomechanical constraints [32,33]. Concerning the characteristics of their bone, rats have a lamellar bone with a limited cancellous architecture and less cortical remodeling [33]. In addition, the bone healing capacity is enhanced in rats [34], and there are significant differences in composition, density, and tissue quality, as well as a distinct locomotion process and biomechanics compared with humans [35]. Rabbits’ bones remodel quickly and also present a different microstructure, compared with humans [33], such as a decreased cancellous bone content [36], smaller-sized long bones, and thin and fragile cortices [37]. Despite the structural differences, the rabbit model has been widely used in bone tissue research, especially due to the availability, ease of housing and handling, and increased trabecular structure, compared with rats [38]. Lastly, large animals seem to be more adequate models for bone-related research, despite the increased cost, housing, and handling requirements, broadly presenting increased research translationally [30]. As an advantage, adult sheep have similar body weights and bone morphology and size closer to that of humans [38]. Further, the rate of bone healing is more approximate to that of humans [17]. Sheep have a similar cancellous/cortical bone organization and undergo bone remodeling while presenting plexiform bone and fewer Haversian canals, substantiating a distinct microstructure [33]. Sheep also report a higher bone density than humans [35]. Accordingly, the transfer of bone research data attained within experimental animal models to the human clinical situation should be carefully conducted, given the distinct morpho-functional characteristics of the small and large animals’ bone tissue [35].

### 4.2. Bone Defect

All studies have used critical-sized bone defects, which are those that will not heal spontaneously during the lifetime of the animal [39]. A critical-size defect will always require a therapeutic approach for complete defect healing, substantiating an increased regenerative requirement [40]. Regarding the establishment of the defect model, all studies used orthotopic defects found in long bones—as the femur [17,21,25] and tibia [19,20,22,23,24,26], as well as the calvarial bone [18]. Briefly, four studies involving the rabbit model selected a tibial location. Among them, two [19,23] used the proximal location, another [20] selected the distal region, and the other study did not report [24]. Among rat models, one study selected the distal femur [21], another study used the proximal tibia [22], one study used the calvarial bone [18], and two studies did not report the bone location [25,26]. The only study in the sheep model selected a femoral location [17]. Predominantly, the proximal tibia comprises cancellous bone, while most of the bone content from the distal tibia, femur, and calvarial is cortical. Cancellous bone has a porous structure and a wider surface, which contribute to faster revascularization for new bone formation, further displaying a bone turnover that is higher than cortical bone [41]. Additionally, it is important to note that rats display the fastest cancellous bone healing, with rabbits presenting an intermediate time and sheep taking the longest healing time [42]. Therefore, attained differences—embracing the anatomical differences, defect size, and healing capabilities—may result in a potential challenge for the comparative analysis of the results.

### 4.3. Bone infection Model

Infection models were all established with the local inoculation of *S. aureus*; a Gram-positive Staphylococcus acknowledged to predominate in the etiology of bone tissue infections [43]. Usually, *S. aureus* is a commensal inhabitant of the skin and mucosal microflora; the ability of *S. aureus* to colonize and cause host infection is possibly due to the production of distinct virulence factors such as adhesins, cytolytic toxins, immune evasion factors, and superantigens [43,44]. *S. aureus* inoculation, in groups without an antibacterial strategy, induced significant tissue destruction [17,18,19,20,21,22,23,24,25,26]. This seems to elapse from a bacteria-dependent decrease in osteoblast activity and enhanced osteoblastic death, impairing proper bone metabolism and formation [4]. In addition, it was also observed that an increased neutrophil infiltration in the presence of *S. aureus* [17,18,21,22,23,24,26] is associated with an abnormal bone healing response [5]. Additionally, the healing process can be compromised by the formation of fibrous tissue, a characteristic of the atrophic non-union [5], as reported in different studies [17,18,25,26] within the implantation area of constructs devoid of the antibacterial strategy. Even upon long-term healing, defects without the presence of the antibacterial agent showed signs of infection, including cortical discontinuity and cortical thickening with distinctive osteolysis [19,21]. Notwithstanding, differences regarding the evolution of bone tissue infection were verified between the different models. They can be broadly attributed to the use of distinct *S. aureus* strains and dissimilar bacteria inoculum, as well as to the distinct bone sites and animal species with dissimilar immune-inflammatory and tissue healing responses.

### 4.4. Scaffolds

Composite systems were found to be the most selected constructs for the healing of infected bone tissue, presumably due to the incorporation of benefits from different materials with dissimilar properties. Commonly, this combination was found to be applied in the form of co-polymers or polymer–ceramic composites [17,18,19,20,21,22,23,24,25,26]. In a system [17], PLGA was used as a biodegradable synthetic polymer. Since the degradation of the polymer particles can be altered by the modification of the lactide:glycolide ratio and the addition of plasticizers, PGE was added to alter the compound’s thermal profile, resulting in the production of thermally sensitive particles [17]. Similarly, another study selected PLLA with the addition of PGE [18], originating the production of porous microspheres with a rough surface, which expectedly favored cell attachment and proliferation [18]. Li et al. [20] used PCL as a biodegradable synthetic polymer associated with PDA, which is a simple method for surface modification of materials. PCL/PDA has a promoted capacity for cell adhesion, proliferation, and osteogenic differentiation [20]. These synthetic polymers [17,18,20] display an increased versatility and manipulation for the enhancement of mechanical properties and control of the degradation rate [45]. However, the drawback of having lower bioactivity and biological interaction with cells, as compared with natural polymers [45]. In our review, only one study used a natural polymer, a chitosan-based hydrogel [24], showcasing the systems’ ability to act simultaneously as a scaffold and a controlled drug delivery system. Overall, the use of composite-based polymers has the potential for improving drug bioavailability, which seems to allow a greater cell adhesion associated with a controlled drug release, providing continuous doses over prolonged periods, increasing drug bioactivity and targeting [46].

Our findings further showed that HA was the most common bioceramic used in the composites’ formulation [21,23,25]. Nevertheless, each study has used it in combination with other materials with different properties, such as collagen [21]—which endorses the system with the ability to control drug diffusion and, in conjunction with HA, presents a “sponge-like” elasticity that makes it easier to fit into bone defects and expectedly enhances the biological outcome [21,47]. Two other studies used HA associated with a PU-based composite [23,25] due to good flexibility, biodegradability, and cytocompatibility. This composition has been shown to promote good biocompatibility and osteoconduction, promoting bone repair without cytotoxicity [23,25]. The most recent study used BG [26] as a biomaterial due to the expected higher quantity and quality of bone formation compared with synthetic HA, probably due to a higher degradation rate [48]. In addition, BG was associated with a polymer matrix composed of PLGA, PEG, and PCL. Lastly, a CaP/CaS composite biomaterial was considered [22], providing an efficacious vehicle for drug delivery and adequate biodegradable and osteogenic capabilities. The incorporation of bioceramic particles in composites for bone applications is expected to enhance tissue regeneration, improving the overall bioactivity of the system [49], further mimicking the composition of bone tissue, which is composed of a combination of inorganic HA crystals and organic collagen fibers [45]. Additionally, ceramics are further intended to increase the mechanical properties of the system [49] and, expectedly, enhance the differentiation of osteoprogenitor cells and the activity of osteoblasts, consequently promoting enhanced regeneration [25].

Only one study reported the use of metals for the bone healing approach in the infected environment [19]. Titanium-based alloys are the most used in structures for bone healing in the infected environment; despite the excellent mechanical strength, good corrosion resistance, and biocompatibility, the disadvantage of non-biodegradability and limited processability in the biological milieu have been reported [19,47].

### 4.5. Antibacterial Strategies

Vancomycin [19,21,22,24,26] and silver (Ag) [18,20,23,25] were the most used antimicrobial strategies in the included studies. However, one study also used cefotiam associated with vancomycin [21], and another used Ag associated with tannin [25]. Currently, there are no consistent guidelines for the management of bone tissue infection. However, antibiotic delivery, either systemically or locally, is the most commonly used therapeutic approach. Vancomycin, a broad-spectrum agent, is highly effective against acknowledged agents associated with bone tissue infection, such as MRSA and other *Staphylococci*, being widely used and typically administered intravenously [50,51]. Moreover, vancomycin has been used in local drug delivery strategies, showing a good penetration profile into the bone [52] and particularly effective in *S. aureus* targeting [22,53]. In local approaches, the properties of the drug carriers may affect the antibacterial efficacy and bone healing capabilities of the system, especially affecting the kinetics of the drug release. Generally, the use of vancomycin has been shown to contribute to bone regeneration by inhibiting bacterial infection [19,21,22,24,26]. When compared with the use of cefotiam, vancomycin showed improved results regarding the eradication of infection, reduction of bone destruction, and regeneration [21]. Egawa et al. [21] reported the combined use of both antibiotics given the high adsorption to the established drug delivery system (HAp/Col). Alternatively, Boyle et al. [22] selected water-soluble vancomycin based on the maintenance of the high drug release after full biomaterial (Cap/Cas) elution. In addition to vancomycin and cefotiam, gentamicin sulfate and clindamycin hydrochloride, which are also commonly used in the treatment of bone infection within cement spacers, due to their thermal stability and hydrophilic essence [54] were used jointly with PLGA/PEG scaffolds(17). This antibiotic combination was used to further reduce the risk of bacterial resistance, potentially arising at the bone site [17].

Despite the overall positive outcome, the release kinetics of the developed systems should be critically evaluated, as high concentrations of antibiotics were previously found to hinder osteoblastic functionality [55] and possibly associated with a delayed healing/regeneration process [5].

Over the past decade, Ag and AgNPs have been widely used as antimicrobial agents due to their antibacterial efficacy [50] and their broad-spectrum and low cytotoxicity properties [56]. One of the main mechanisms of antimicrobial action is the disturbance of the respiratory chain affecting energy generation [57], seconded by the induction of structural changes in bacterial cells, able to modulate the production of intracellular reactive oxygen species [58]. More recently, AgNPs have attracted much attention among researchers. AgNPs seem to display an increased functionality, possibly due to the increased active surface area [59]. It is interesting to note that this activity is strongly dependent on the nanoparticles’ size, with smaller nanoparticles displaying an enhanced capacity to penetrate cells—either eukaryotic or prokaryotic, establishing an increased concentration and, consequently, an increased toxic activity [50,60,61]. In this regard, a high dosage of AgNPs showed significant cytotoxicity followed by cell death within eukaryotic populations. However, lower dosages maintained the effective antibacterial activity without interfering with the osteoblastic cell functionality [18,20]. Overall, studies considering Ag or AgNPs supported an effective and significant outcome regarding the control of bone infection, compared with the control group [18,20,23,25].

Lastly, a study reported the use of tannin-mediated Ag-NPs-coated HA (Ag-THA) and its combination with polyurethane to develop a new antibacterial system for bone applications [25]. Tannin is known to exhibit considerable antimicrobial activity, especially against Gram-positive bacteria such as *S. aureus* [62,63]. The combination of tannin with Ag-NPs was found to be more effective in bacterial management than the implantation of AgNP-loaded scaffolds devoid of tannin in the present model [25].

## 5. Conclusions

This work revealed that in all reviewed studies, a significant improvement in the healing of the infected bone was achieved when an antibacterial strategy was considered within the implanted construct, particularly at longer time points of analysis. Given the wide variability of constructs, data suggests that regardless of the antibacterial strategy or the construct composition, success was achieved in the resolution of tissue infection, further entailing the promotion of bone healing. Nonetheless, the wide diversity of experimental protocols precluded the meta-analytical evaluation of the data. Lastly, considering the assessment of bias, most included studies revealed an inadequate reporting methodology, which may lead to an unclear or high risk of bias and directly hinder future studies. Therefore, these findings should be taken with caution. Future research should improve the quality of the reports through the standardization of animal models, sample size, bone type, and study design, focusing on issues such as randomization and blindness, outcome measures, and detailed data reporting. Better conduct on the report of experimental animal model data may be obtained through guidelines such as Reporting In Vivo Experiments (ARRIVE).

## Figures and Tables

**Figure 1 jfb-13-00193-f001:**
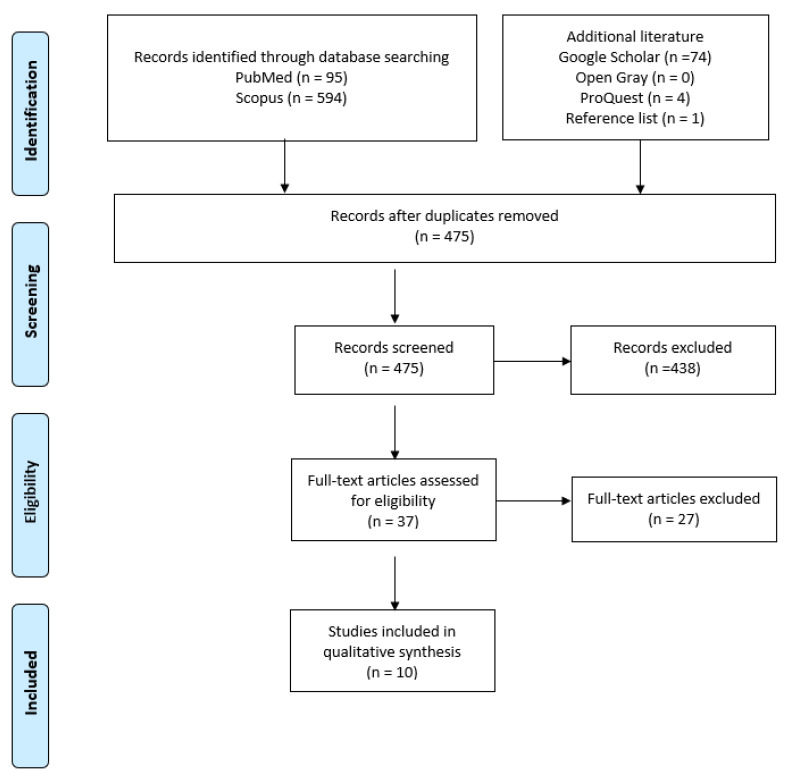
PRISMA diagram. Flow diagram of the systematic review literature search results. Based on ”Preferred Reporting Items for Systematic Reviews and Meta-Analyses: The PRISMA Statement.”

**Figure 2 jfb-13-00193-f002:**
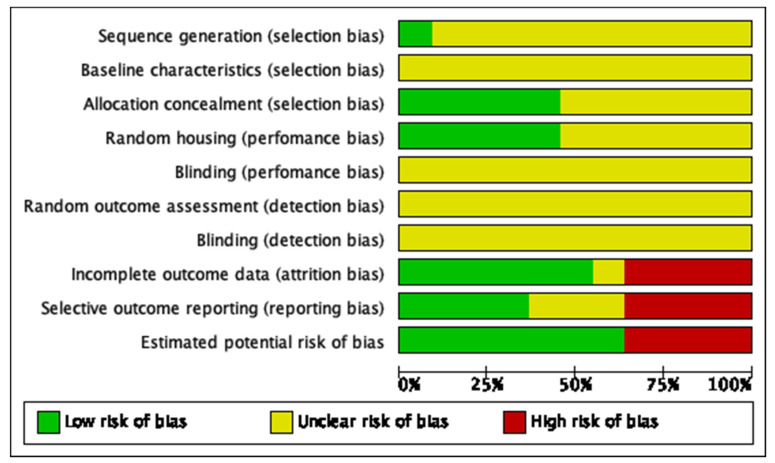
The risk of bias of each item of the SYRCLE tool, for the included studies. The RevMan Version 5.3 (The Nordic Cochrane Centre, The Cochrane Collaboration, Copenhagen, 2014) software was used.

**Table 1 jfb-13-00193-t001:** Description of the main characteristics of the studies included in the analysis.

Author and Year	Specie	Total Number of Animals	Bone Defect Type and Location	Biomaterial/Scaffold	Antibacterial Agent	Bacteria	Strain	Inoculum Size	Time to Insert the Scaffold
McLaren et al., 2014 [17]	Sheep	30	Ø: 8 mm and height: 4 mm Medial femoral condyle	PLGA/PEG	*Gentamicin sulfate and Clindamycin hydrochloride*	*S. aureus*	F2789	20 μL of a 2 × 10^6^ cfu/mL (40,000 CFU)	At the same time as inoculation
Wei PF et al., 2019 [18]	Rats	45	Ø: 8 mm in the cranium	PLLA/PEG	*Silver nanoparticles (AgN*Ps)	*S. aureus*	Not reported	1 × 107 CFU in 100 μL of sterile normal saline	1 week
Li et al., 2019 [20]	Rabbits	16	Ø: 4 mm and height: 4 mm External tibial epicondyle of left limb.	PCL	*Silver nanoparticles (AgN*Ps)	*S. aureus*	ATCC 25923	0.1 mL of 1 × 10^5^ CFU/mL	At the same time as inoculation
Boyle et al., 2019 [22]	Rats	64	Ø: 3 mm and height: 3 mm Right proximal tibia	Cap/Cas	*Vancomycin (10%)*	*S. aureus*	ATCC 29213	10 μL of 1.5 × 10^6^ CFU/ml	At the same time of inoculation and 3 weeks after inoculation
Zhang et al., 2019 [23]	Rabbits	54	A cortical bone window of 10 × 6 mm the proximal tibia.	Ag/n-HA/PU	*Silver* phosphate	*S. aureus*	ATCC 25923	0.1 mL of 3 × 10^7^ CFU/mL	4 weeks
Zhang et al., 2020 [19]	Rabbits	18	Ø: 5 mm tibial plateau	TS-M/P/V	*Vancomycin*	MRSA	Not reported	1 × 10^8^ CFU	At the same time as inoculation
Egawa et al., 2020 [21]	Rats	54	Ø: 1 mm hole made in the first surgery was dilated to Ø: 3 mm Lateral epicondyle of the bilateral femur.	HAp/Col	Cefotiam and Vancomycin	MSSA	Not reported	1 × 10^7^ CFU	1 week
Jin Tao et al., 2020 [24]	Rabbits		Ø: 2 mm Tibia	VCM-NPs/Gel	*Vancomycin*	*S. aureus*	ATCC 96	1 × 10^8^ CFU/mL	4 weeks
Tian et al., 2021 [25]	Rats	60	Ø: 3.5 mm × 5 mm the lateral condyle of the femur	PU/Ag-THA HA/PU	silver nitrate (AgNO_3_) and Tannin (T)	*S. aureus*	Not reported	1 × 10^6^ CFUs/mL	10 days
Hasan et al., 2021 [26]	Rats	6	A 4.2 mm hole the tibial metaphysis.	ABVF-BG	*Vancomycin*	*S. aureus*	ATC49230	1 × 10^8^ CFUs	At the same time as inoculation

**Table 2 jfb-13-00193-t002:** Description of the included studies outcomes: bone volume fraction (BV/TV).

BV/TV		
Study	Time point	Outcome
Wei PF et al., 2018 [18]	24 weeks	Antimicrobial strategies: 47.5 ± 1.39% mg cm^−3^ and Control group: 52 ± 0.99% mg cm^−3^ (*p* < 0.01)
Li et al., 2019 [20]	8 weeks	Antimicrobial strategies showed significantly higher values of BV/TV in PCL/AgNPs and PCL/PDA/AgNPs groups, compared with other groups, with the highest levels on the PCL/PDA/AgNPs (*p* < 0.05)
Boyle et al., 2019 [22]	6 weeks	Inoculation + CaS/CaP (at the same time) group: 20.55% Inoculation + after 3 weeks: CERAMENT + Vancomycin group: 31.27% (*p* < 0.05)
Zhang et al., 2019 [23]	4 days	Almost no new bone formation was seen in any group
	3 weeks	The new bone formation was observed in each group, the n-HA/PU3 group (0.113 ± 0.047) showed the most obvious change, but there were no significant differences
	6 weeks	Levels were increased in each group, and there was a significant difference between the n-HA/PU3 group (0.488 ± 0.100) and n-HA/PU group (0.131 ± 0.064; *p* = 0.01)
	12 weeks	Levels were increased in the n-HA/PU3 group (0.607 ± 0.043), as well as in the n-HA/PU10 group (0.636 ± 0.088), and both the n-HA/PU3 group and n-HA/PU10 group showed significantly higher bone formation than the n-HA/PU group (0.057 ± 0.057). There were no significant differences in the rate of bone formation between the n-HA/PU3 group and the n-HA/PU10 group
Jin Tao et al., 2020 [24]	8 weeks	The BV/TV decreased further and was significantly lower in the control group than in the VCM/Gel, VCM-NPs/Gel, and VCS groups In addition, the BV/TV was markedly higher in the VCMNPs/Gel group than in the VCM/Gel group

**Table 3 jfb-13-00193-t003:** Description of the included studies outcomes: bone mineral density (BMD).

BMD		
Study	Time Points	Outcome
Wei PF et al., 2018 [18]	24 weeks	Antimicrobial strategies: 359.05 ± 30.99 mg cm^−3^ and Control 65.41 ± 11.21 mg cm^−3^ showing significant differences (*p* < 0.01)
Jin Tao et al., 2020 [24]	8 weeks	The BMD decreased further and was significantly lower in the control group than in the VCM/Gel, VCM-NPs/Gel, and VCS groups. In addition, the BMD was markedly higher in the VCMNPs/Gel group than in the VCM/Gel group
Tian et al., 2021 [25]	4 weeks 8 weeks 12 weeks	The PU/Ag-THA group, at all three-time points, presented significantly higher results than that of PU/HA and PU/THA groups

**Table 4 jfb-13-00193-t004:** Description of the included studies outcomes: cortical bone destruction.

TV		
Study	Time Points	Outcome
Egawa et al., 2020 [21]	1 week	The VCM group (0.45 ± 0.04 mm) showed a significant decrease in cortical destruction compared with the NS group (0.37 ± 0.04 mm) and the same trend was seen in the CEZ group (0.44 ± 0.06 mm), though this difference was not statistically significant
	2 weeks	Both the VCM (0.51 ± 0.07 mm) and CEZ (0.50 ± 0.04 mm) groups showed a marked decrease in cortical bone destruction. NS group showed 0.18 ± 0.07 mm of cortical destruction
	4 weeks	A significant decrease in the treatment group with vancomycin was attained, compared with the control group. The same trend was noted in the cefotiam group, but the difference was not significant

**Table 5 jfb-13-00193-t005:** Description of the included studies outcomes: bone area.

Bone Area		
Study	Time Points	Outcome
Boyle et al., 2019 [22]	6 weeks	An identical value of the bone area was attained for the antibacterial biomaterial and control group (1.46 mm^2^)

**Table 6 jfb-13-00193-t006:** Description of the included studies outcomes: trabecular number (Tb.N).

Tb.N		
Study	Time Points	Outcome
Li et al., 2019 [20]	8 weeks	The ratio of Tb.N was significantly higher in PCL/AgNPs and PCL/PDA/AgNPs groups, as compared with other groups, with the PCL/PDA/AgNPs group achieving the highest score
Zhang et al., 2019 [23]	4 days	There were a few new bone trabeculas
	3 weeks	The n-HA/PU10 group (0.264 ± 0.139/mm) showed the highest number of new bone trabeculas, but without significant differences
	6 weeks	The number of new bone trabeculas increased in each group, and a significant difference was detected between the n-HA/PU3 group (1.486 ± 0.129/mm) and n-HA/PU group (0.621 ± 0.256/mm; *p* = 0.013)
	12 weeks	The number of new bone trabeculas continued to increase in all groups, but there was no difference between the n-HA/PU3 group (1.595 ± 0.319/mm) and n-HA/PU10 group (1.711 ± 0.379/mm)

**Table 7 jfb-13-00193-t007:** Description of the included studies outcomes: Trabecular thickness (Tb.Th).

Tb.Th		
Study	Time Points	Outcome
McLaren et al., 2014 [17]	13 weeks	Antimicrobial strategies group showed a bone defect fill of 53.8 ± 17.2%, but no significant differences were attained, compared with the control
Zhang et al., 2019 [23]	4 days	There were no significant differences in the thickness of the new bone trabecular thickness among all groups
	3 weeks	n-HA/PU3 group = 0.142 ± 0.031 μm and Control group: 0.029 ± 0.001 μm with statistical significance (*p* = 0.027)
	6 weeks	n-HA/PU3 group = 0.327 ± 0.040 μm and Control group: 0.067 ± 0.026 μm with statistical significance (*p* = 0.035)
	12 weeks	n-HA/PU3 group = 0.140 ± 0.095 μm and Control group: 0.157 ± 0.065 μm with statistical significance (*p* = 0.002)
Hasan et al., 2021 [26]	8 weeks	A healing bone without signs of infection was seen in the antimicrobial strategies group. Precisely, the drilled hole at the bone site was being filled by immature cancellous and cortical bone. However, the numerical data were not reported, neither the statistical significance of the result

**Table 8 jfb-13-00193-t008:** Description of the included studies outcomes: Trabecular separation (Tb.Sp).

Tb.Sp		
Study	Time Points	Outcome
Zhang et al., 2019 [23]	3 weeks	There were no significant differences between any of the groups.
	6 weeks	Ag 3%: 0.351 ± 0.099 μm and control group 1.245 ± 0.276 μm with statistical significance (*p* = 0.013).
	12 weeks	Ag 3%: 0.250 ± 0.022 μm and control group 0.507 ± 0.122 μm with statistical significance (*p* = 0.042). Ag 10%: 0.218 ± 0.055 μm and control group 0.507 ± 0.122 μm with statistical significance (*p* = 0.038).

**Table 9 jfb-13-00193-t009:** Risk of bias assessment for included articles (SYRCLEs).

Criteria [16]	McLaren et al., 2014 [17]	Wei et al., 2018 [18]	Li et al., 2019 [20]	Boyle et al., 2019 [22]	Zhang et al., 2019 [23]	Zhang et al., 2020 [19]	Egawa et al., 2020 [21]	Jin Tao et al., 2020 [24]	Tian et al., 2021 [25]	Hasan et al., 2021 [26]
Sequence generation (selection bias)	Unclear	Unclear	Unclear	Unclear	Unclear	Unclear	Unclear	Unclear	Unclear	Unclear
Baseline characteristics (selection bias)	Unclear	Unclear	Unclear	Unclear	Unclear	Unclear	Unclear	Unclear	Unclear	Unclear
Allocation concealment (selection bias)	Low risk	Low risk	Low risk	Low risk	Unclear	Low risk	Unclear	Unclear	Unclear	Unclear
Random housing (performance bias)	Low risk	Unclear	Unclear	Low risk	Unclear	Unclear	Low risk	Low risk	Unclear	Unclear
Blinding (performance bias)	Unclear	Unclear	Unclear	Unclear	Unclear	Unclear	Unclear	Unclear	Unclear	Unclear
Random outcome assessment (detection bias)	Unclear	Unclear	Unclear	Unclear	Unclear	Unclear	Unclear	Unclear	Unclear	Unclear
Blinding (detection bias)	Unclear	Unclear	Unclear	Unclear	Unclear	Unclear	Unclear	Unclear	Unclear	Unclear
Incomplete outcome data (attrition bias)	Low risk	High risk	Low risk	Low risk	High risk	Low risk	High risk	High risk	High risk	Unclear
Selective outcome reporting (reporting bias)	Low risk	High risk	High risk	Unclear	Low risk	High risk	Low risk	High risk	Low risk	Unclear
Estimated potential risk of bias	Low risk	High risk	High risk	High risk	Low risk	Low risk	Low risk	Low risk	Low risk	High risk

## Data Availability

The data presented in this study are available on request from the corresponding author.

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
