# Peer review of "Antimicrobial Biomaterials for the Healing of Infected Bone Tissue: A Systematic Review of Microtomographic Data on Experimental Animal Models"

_jfb, 2022, doi:10.3390/jfb13040193_

Round 1

Reviewer 1 Report

This manuscript is a systematic review on the use of antimicrobial biomaterials for the healing of infected bone tissue on animal models evaluated by micro-CT. The review protocol was based on PRISMA guidelines and is registered in PROSPERO (#CRD42020197148). Additionally, the risk of bias was assessed by SYRCLE’s tool. The initial number of articles after duplicates were removed was 475. Out of them, the review was based on 10 studies. The authors accurately concluded that there is a clear improvement in bone healing of infected bone when an antibacterial strategy was considered within the implant construct, particularly at longer time points. Agreeing with the authors, this reviewer did not find another published systematic review on this subject in PubMed, which confirms the originality of the study. The text is well constructed and written, and before publication it is suggested a final check of the English language. Finally, this reviewer did not use any tool to check for plagiarism. Thus, to preserve the authors and the Journal, it will be important to do so, if it has not already been done.

Author Response

The authors acknowledge the reviewer's comments and inputs. Accordingly to the suggestion, the text was revised for English language and style.

Reviewer 2 Report

1.      Please check the font. It changes throughout the document.

2.      As pointed out in the discussion, there is a very high degree of heterogeneity among the experimental protocols, as for animal species, study design, different bone implantation locations, type and composition of scaffolds and antibacterial strategies, measurement methods and outcome criteria. It would be beneficial to collect data from similar studies for comparison in critical parameters at least in animal species, antibacterial strategies, measurement methods and outcome criteria.

3.      The results with and without antibacterial strategy is incomparable. This comparison cannot be used to draw any useful conclusions.

Author Response

The authors acknowledge the reviewer's comments and the constructive feedback. Detailed answers to the specific topics are following detail:

  1. Please check the font. It changes throughout the document.

The font and the overall formatting of the manuscript were corrected.

  1. As pointed out in the discussion, there is a very high degree of heterogeneity among the experimental protocols, as for animal species, study design, different bone implantation locations, type and composition of scaffolds and antibacterial strategies, measurement methods and outcome criteria. It would be beneficial to collect data from similar studies for comparison in critical parameters at least in animal species, antibacterial strategies, measurement methods and outcome criteria.

In accordance with the reviewer's comment, a high degree of heterogeneity was identified in the published studies, which precluded a meta-analytical approach as initially intended. Accordingly, by focusing on microtomographic data, we aimed to standardize measurement methods, setting this as the ground base for the review. The data collection and analysis within more similar studies may be a very interesting research approach for future studies, aiming to focus more exclusively on animal species or antibacterial studies, as suggested. With this review, and given the absence of previous works on the topic, we aimed to establish a first approach to the theme entailing a more global methodologic review strategy.  

  1. The results with and without antibacterial strategy is incomparable. This comparison cannot be used to draw any useful conclusions.

From a biological point of view, tissue healing differs significantly in the presence or absence of infection and, as well, in the presence or absence of an antibacterial strategy. Nonetheless, this review was focused on the assessment of the efficacy of antibacterial strategies in the healing of infected bone tissue. Accordingly, and to validate the data, the inclusion of experimental groups with materials without the antibacterial strategy was mandatory.

Reviewer 3 Report

The authors prepared a valuable review entitled "Antimicrobial biomaterials for the healing of infected bone tissue: a systematic review of microtomographic data on experimental animal models" they covered the essential data in their work. This study is well-organized and comprehensively described. Additionally, this research is scientifically sound and not misleading. I suggest publishing this work in its current form.

Reviewer 4 Report

The manuscript entitled “Antimicrobial biomaterials for the healing of infected bone tis- 2 sue: a systematic review of microtomographic data on experi- 3 mental animal models” provides a very interesting systematic review on a very hot topic. The manuscript is well written and well structured. However, several points must be addressed in order to be suitable for publication. Thus, important reviews on this topic such as https://doi.org/10.1016/j.mtbio.2022.100412 should be included in the introduction section of this review. Besides, the novelty of this review with respect to previous reviews on this topic must be highlighted in the abstract and introduction section.

The last paragraph of the introduction section is quite confusing.

Author Response

The authors acknowledge the reviewer's comments and the constructive feedback. Detailed answers to the specific topics are following detail:

The manuscript entitled “Antimicrobial biomaterials for the healing of infected bone tissue: a systematic review of microtomographic data on experimental animal models” provides a very interesting systematic review on a very hot topic. The manuscript is well written and well structured. However, several points must be addressed in order to be suitable for publication.

Thus, important reviews on this topic such as https://doi.org/10.1016/j.mtbio.2022.100412 should be included in the introduction section of this review.

  • We acknowledge the reviewer for the suggestion. The referred work was cited accordingly, in the Introduction section.

Besides, the novelty of this review with respect to previous reviews on this topic must be highlighted in the abstract and introduction section.

  • The abstract and Introduction section were modified to further reflect the novelty of the manuscript.

The last paragraph of the introduction section is quite confusing.

  • The last paragraph of the Introduction section was modified according to the suggestion.